# "I'm in the Bluesky Tonight": Insights from a year worth of social data

**Andrea Failla**[1,2]*, **Giulio Rossetti**[2]

**1** Department of Computer Science, University of Pisa, Pisa, Italy, **2** National Research Council, Institute of Information Science and Technologies "A. Faedo" (ISTI), Pisa, Italy

* andrea.failla@phd.unipi.it

## Abstract

Pollution of online social spaces caused by rampaging d/misinformation is a growing societal concern. However, recent decisions to reduce access to social media APIs are causing a shortage of publicly available, recent, social media data, thus hindering the advancement of computational social science as a whole. We present a large, high-coverage dataset of social interactions and user-generated content from Bluesky Social to address this pressing issue. The dataset contains the complete post history of over 4M users (81% of all registered accounts), totalling 235M posts. We also make available social data covering follow, comment, repost, and quote interactions. Since Bluesky allows users to create and like *feed generators* (i.e., content recommendation algorithms), we also release the full output of several popular algorithms available on the platform, along with their timestamped "like" interactions. This dataset allows novel analysis of online behavior and human-machine engagement patterns. Notably, it provides ground-truth data for studying the effects of content exposure and self-selection and performing content virality and diffusion analysis.

## Introduction

Online social platforms (OSPs) have traditionally been a conspicuous data source for studying online human behaviors. From discourse analysis [1, 2], to studying d/misinformation [3, 4], coordinated behaviors [5, 6], radicalization [7, 8], echo-chamber effects [9, 10], and critical event detection [11, 12], computational social science has long laid its development on—and answered its question through—social media data. However, in early 2023, Twitter/X and Reddit announced their plans to discontinue free access to their API services. These decisions have slowed down the advancement of computational social science research. Indeed, in the *post-api era* [13], researchers are left with limited options when it comes to finding suitable data. One option is to exploit custom web scrapers, i.e., programs that pretend to be regular users and collect data as they navigate OSPs. This strategy, however, is incredibly time-consuming and often violates the platforms' terms of use. Moreover, it is hardly reusable, as a change in the website's source code may easily break the scraper. Another possibility is to use search engines to query for specific OSPs and collect the result pages via the engine's API. In other words, one may circumvent an OSP's API restrictions by collecting data via a (often more

**Data Availability Statement:** All data and code are available in a dedicated zenodo repository at https://zenodo.org/records/11082879.

**Funding:** This work is supported by (i) the European Union – Horizon 2020 Program under the scheme "INFRAIA-01-2018-2019 – Integrating

Activities for Advanced Communities", Grant Agreement n.871042, "SoBigData++: European Integrated Infrastructure for Social Mining and Big Data Analytics" (url{http://www.sobigdata.eu}); (ii) SoBigData.it which receives funding from the European Union – NextGenerationEU – National Recovery and Resilience Plan (Piano Nazionale di Ripresa e Resilienza, PNRR) – Project: "SoBigData. it – Strengthening the Italian RI for Social Mining and Big Data Analytics" – Prot. IR0000013 – Avviso n. 3264 del 28/12/2021; (iii) EU NextGenerationEU programme under the funding schemes PNRR-PE-AI FAIR (Future Artificial Intelligence Research). The funders had no role in study design, data collection and analysis, decision to publish, or preparation of the manuscript.

**Competing interests:** The authors have declared that no competing interests exist.

permissive) search engine API. This strategy is less costly but was shown to be strongly biased in favor of (i) popular social media users, (ii) positive content, and (iii) non-political content [14]. The last option is to rely on older datasets. However, these quickly become outdated, let alone reusing a specific data sample, which introduces inherent biases and prevents the results' generalization. Moreover, many OSPs (including Twitter/X) only allow sharing identifiers of social media posts, which must be used to obtain complete post metadata via the OSP's API—which is now prohibitive.

Contrary to the general trend, however, the year-old decentralized OSP *Bluesky Social* (hereafter, *Bluesky*) has recently opened its APIs to developers, offering a potential solution to the widespread data shortage. Additionally, Bluesky offers unique features that make it a valuable resource for new studies, such as a new open federation technology—the AT protocol [15]—and augmented algorithmic choice. Regarding this last point, Bluesky allows users to create and like custom content recommendation algorithms called *feed generators* [16]. This feature opens up new ways in which human-AI relationships can be investigated, naturally favoring interesting research questions, e.g., does choosing your algorithm increase/decrease the risk of opinion polarization and of coming across d/misinformation? How do custom feeds relate to/affect the identification of reliable information sources? With this work, we aim to start bridging these gaps with a manifold contribution. First, we introduce a curated Bluesky dataset comprising more than 4M accounts ($\sim$81% of all registered users according to the latest information available [17]) along with their complete posting activity ($\sim$235M posts) and follower/followee relations; as part of this dataset, we also release the output of 11 feed generator algorithms available on the platform (i.e., the full collection of posts retrieved by such algorithms), along with data on who liked such posts and when. The dataset is complemented by Python scripts implementing the data collection and processing pipelines, potentially allowing other researchers to collect/process more data according to their needs; moreover, we promote a preliminary descriptive analysis of Bluesky's structure, dynamics, and content. Aside from the work introducing the AT protocol [15] and a study on migration from Twitter to other OSPs [18], ours is the first work focusing on Bluesky Social, and the data is likely to be one of the highest-coverage datasets on online social platforms.

## Bluesky social

At launch (February 17th, 2023), the Bluesky service was available to new users only under invitation from an existing user or Bluesky PBC. The beta program has had moderate success since, capturing a considerable slice of ex-Twitter/X users during the 2023 mass migration [18]. Despite the launch of Meta's much more popular competitor, Threads, in the same year [19], Bluesky reached more than three million users in November. Moreover, following the removal of the invite-only policy, the platform has reported an unprecedented increase in new user activity, totalling 5 million users in February 2024. Regarding user experience, Bluesky is comparable to more established microblogging services like X and Mastodon. Fig 1 (left) displays a typical Bluesky screen. Users can post short-form content, such as texts up to 300 characters and up to four images. Posts may also contain links to external websites, mentions to other users, and be tagged via hashtags. On creation, posts with attached media can be labelled with tags that advise viewer discretion (e.g., adult content warnings). Like most social media sites, users can interact with other posts by replying, sharing, or liking. Bluesky distinguishes between *reposting*, i.e., sharing another user's post as is, and *quoting*, i.e., reposting and adding a comment. The follower/following feature is implemented in a directed fashion, meaning user A befriending user B does not imply B befriending A. Perhaps the most distinctive feature of Bluesky is its *feeds* functionality. The service allows users to choose the algorithm(s) that

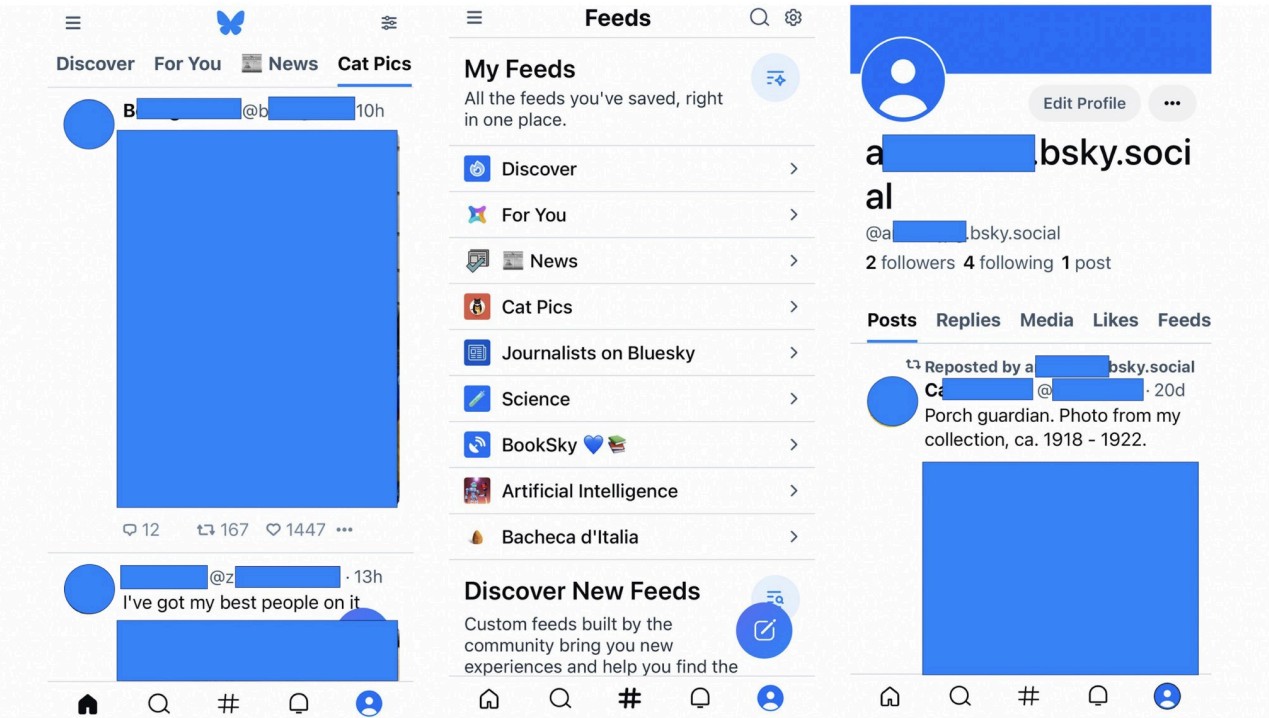

**Fig 1.** Screenshots of the *home* (left), *feeds* (middle), and *profile* (right) tabs from Bluesky's official iOS app (v1.71). In the home tab, the top row is a scrollable bar listing the user's bookmarked feed generators. The post at the top only contains an image and received 12 comments, 167 reposts, and 1447 likes. The post at the bottom contains both text and an image. The *feeds* tab contains the list of bookmarked feed generators, along with a feed search bar. Finally, the *user* tab shows the logged-in user's profile. Shared pictures were obfuscated to avoid sharing potentially copyrighted material. All panels were captured by the author on his device.

power their home feed, allowing them to view, e.g., only posts by whom they follow, posts containing specific words or entities, and more complex filters. Feeds are implemented in a way that lets the user subscribe to multiple feeds and effortlessly switch from one to another (see Fig 1 (middle)). Apart from the default `Following` feed (not shown), yielding content from followed users in reverse-chronological order, the user in Fig 1 is subscribed to the `Discover` and `For You` feeds, which yield trending content. These aim to mimic the standard feeds of other platforms, such as TikTok or Instagram. The logged-in user is also subscribed to a `News` feed that shows [h]*eadlines from verified news organisations* [20] as well as the `Cat Pics` feed (on display), yielding posts with cat pictures [21]. While the former three were created by Bluesky developers, the latter are created by other users: indeed, Bluesky also offers the possibility to build and share new feed algorithms via freely available software tools.

## Materials and methods

### Data collection

We collected publicly available user data from Bluesky using its official Developer API [22]. The process consisted of three phases. We collected the users' followers in the first phase (from February 25th to March 2nd, 2024). We obtained its list of followers from Bluesky's official account `@bsky.app`. With a breath-first approach, we repeated this procedure for every found user until no new user was encountered. Once a sample of 1M unique users was collected, we distributed subsequent requests among ten machines to reduce collection time.

In the second phase (from March 19th to March 21st, 2024), we parsed the collected users to obtain their following lists (that is, for each user, the list of accounts she follows) and thus improve coverage. The procedure resulted in a sample of 4,099,699 unique users. Based on the latest publicly available official information [17], our sample covers $\sim 81\%$ of Bluesky accounts. In this phase, we also collected posts from user timelines and feeds. We collected all posts from users in our sample via the dedicated API, totalling 237,121,706 posts. In this phase, additional processing was required to identify "reposts". Indeed, while quotes and replies can be easily identified because the original post is clearly referenced in metadata, to the best of our knowledge, it is impossible to tell whether an item is a repost from metadata alone. The only clue is the post's author: if while collecting posts of user *A* we found a post by user *B*, we labelled this as a repost. Unfortunately, we cannot capture self-reposts, although these might be irrelevant depending on the context. Moreover, Bluesky allows self-reposting the same post only once, which might imply a low number of self-reposts in our dataset. We also collected posts from various topics, including science, news, and social issues, that appeared in specific popular feeds. These feeds were manually selected among the most popular feeds as of March 18th.

Finally, in the third phase (April 23th-25th), we collected likes to both the feeds (i.e., whether and when a user liked a feed generator) and its posts (i.e., whether and when a user liked a post yielded by a feed generator. In total, we obtained information on 18,324 feed generator likes and 4,895,318 post likes. To ensure completeness, we waited one month before obtaining "like" interactions, since posts lose their impact after a while [23], thus it is unlikely they will receive many new likes.

## Data processing

The data collection pipeline produced the following outputs: (i) a collection of files containing, for each user, her followers, (ii) a collection of files containing, for each user, her followees, (iii) a collection of files containing user posts, each post represented as a JSON-formatted line, (iv) a collection of files containing posts appearing in specific feeds, each post represented as a JSON-formatted line, (v) information on who liked posts appearing in specific feeds and when, and (vi) information about who liked specific feed generators and when. Collections (i) and (ii) were aggregated into a single file. User handles were pseudo-anonymized during this process by assigning a progressively increasing integer value.

Collections (iii) and (iv) required several processing steps; first, we filtered out post metadata such as entities (e.g., hashtags and mentions) and attached media. Secondly, we filtered out posts with incorrect or ill-formatted timestamps. Indeed, Bluesky allows users and third-party applications to change a post's creation date via the developer API. While this flexibility is, in principle, beneficial—as it allows, for instance, importing posts from other sites (e.g., Twitter/X) or moving content between Bluesky servers— it also has potentially undesirable side effects, such as incorrect date formats and fake dates. We retained only posts between February 17th, 2023, and March 18th, 2024 (inclusive) and limited timestamps to the minute information. Keeping posts published until March 18th (and not later) was done as an attempt to reduce potential disalignments between relational (i.e., followers) and post data; in doing so, we ensure a degree of alignment by considering posts produced up until the final parse on the follower network (March 19th). Thirdly, we obtained each user's instance by relying on handles. Bluesky handles are web domains of the form *un.sd* where *un* is a unique username chosen at creation that identifies an account within the network, and *sd* is the server domain, i.e., of the server hosting that user's data. The server domain may contain dots (see Fig 1, rightmost). We used regular expressions to extract server domains. Then, we mapped user handles

**Table 1. Post metadata.**

| Category | Field | type | #non-null |
|---|---|:---:|:---:|
| User | user_id | int | 235,567,116 |
| | instance | str | 235,567,116 |
| Content | post_id | int | 235,567,116 |
| | date | int | 235,567,116 |
| | text | str | 235,567,116 |
| | langs | list | 220,628,598 |
| | labels | list | 4,027,096 |
| | like_count | int | 235,567,116 |
| | reply_count | int | 235,567,116 |
| | repost_count | int | 235,567,116 |
| | sent_label | int | 128,664,788 |
| | sent_score | float | 128,664,788 |
| Relational | reply_to | int | 87,704,964 |
| | replied_author | int | 87,704,964 |
| | thread_root | int | 87,704,964 |
| | thread_root_author | int | 87,704,964 |
| | repost_from | int | 63,549,643 |
| | reposted_author | int | 63,549,643 |
| | quotes | int | 12,110,474 |
| | quoted_author | int | 12,110,474 |

(both in the corresponding field and the posts' text) with the integer values obtained when processing collections (i) and (ii) and also assigned a unique ID to each post. Moreover, since language metadata is often inconsistent (that is, a post in English may be labelled as english, en, eng, or other variations, e.g., capitalization, incorrect spelling, etc.), we manually mapped language metadata to the ISO 639-2 standard. In this case, we also found multiple occurrences of ill-formatted/non-standard language metadata (e.g., posts labelled as not-a-lang, whatever). In these cases, we kept the post in the dataset but removed the language tag. After these steps, 235,567,116 posts are left in the dataset. Finally, to further enrich the dataset and favor future analysis, we estimated the sentiment of English posts leveraging a case-sensitive RoBERTa model fine-tuned on $\sim$129M English tweets [24]. To reduce noise, since a post can be tagged with multiple languages, we classified only those tagged as English and where no other language appears. As a result, we extended the dataset with the sentiment labels (0: negative, 1: neutral, 2: positive), and the model's confidence scores rounded to the third decimal place. Please refer to Tables 1 and 2 for further information on post metadata. Finally, usernames and posts in collections (v) and (vi) were also assigned pseudonymized IDs and temporal information was aggregated at the minute level.

## Data records

We make our dataset available for further research by releasing it publicly on Zenodo [25] (DOI: https://doi.org/10.5281/zenodo.11082878). To guarantee the reproducibility and transparency of the research, all the code used to collect and clean the data is included in the repository, together with the code to reproduce the experiments. In summary, the Zenodo repository contains the following files:

**Table 2. Post metadata description.**

| Category | Field | description |
|---|---|---|
| User | user_id | an identifier univocally associated with each author/user. |
| | instance | the name of the instance that the user is registered to |
| Content | post_id | an identifier univocally associated with each post |
| | date | the post date and time formatted as YYYYmmddhhMM. |
| | text | the post's text content |
| | langs | ISO 639-2 language codes |
| | labels | the content warning label(s) that the post is tagged with |
| | like_count | the number of likes as per the post metadata |
| | reply_count | the number of replies as per the post metadata |
| | repost_count | the number of reposts as per the post metadata |
| | sent_label | the text's sentiment |
| | sent_score | the sentiment model's confidence |
| Relational | reply_to | the ID of the post to which the current post replies to |
| | replied_author | the ID of the replied post's author |
| | thread_root | the ID of the post that initiated the discussion thread |
| | thread_root_author | the ID of the root post's author |
| | repost_from | the ID of the reposted post |
| | reposted_author | the ID of the reposted post's author |
| | quotes | the ID of the quoted post |
| | quoted_author | the ID of the quoted post's author |

- `followers.csv.gz`. This compressed file contains the anonymized follower edge list. Once decompressed, each row consists of two comma-separated integers *u*, *v*, representing a directed following relation (i.e., user *u* follows user *v*).

- `posts.tar.gz`. This compressed folder contains data on the individual posts collected. Decompressing this file results in 100 files, each containing the full posts of up to 50,000 users. Each post is stored as a JSON-formatted line (see Table 2 for details on specific fields);

- `interactions.csv.gz`. This compressed file contains the anonymized interactions edge list. Once decompressed, each row consists of six comma-separated integers representing a comment, repost, or quote interaction. These integers correspond to the following fields, in this order: `user_id`, `replied_author`, `thread_root_author`, `reposted_author`, `quoted_author`, and `date` (see Table 1). At least one of `replied_author`, `thread_root_author`, `reposted_author`, and `quoted_author` is non-null for all rows;

- `graphs.tar.gz`. This compressed folder contains edge list files for the graphs emerging from reposts, quotes, and replies. Each interaction is timestamped. The folder also contains timestamped higher-order [26] interactions emerging from discussion threads, each containing all users participating in a thread.

- `feed_posts.tar.gz`. This compressed folder contains posts that appear in 11 thematic feeds. Decompressing this folder results in 11 files containing posts from one feed each. Posts are stored as a JSON-formatted line. Fields correspond to those in Table 1, except for those related to sentiment analysis (`sent_label`, `sent_score`), and reposts (`repost_from`, `reposted_author`);

**Table 3. Collected feed statistics.**

| name | #posts | #authors | #post likes | #feed likes |
|---|---|---|---|---|
| #Disability | 566 | 411 | 4,657 | 1,244 |
| #UkrainianView | 2,098 | 172 | 52,308 | 1,026 |
| AcademicSky | 913 | 352 | 4,344 | 803 |
| BlackSky | 86,490 | 1,564 | 1,714,160 | 2,590 |
| BookSky | 738 | 275 | 4,117 | 1,813 |
| Game Dev | 635 | 504 | 4,736 | 1,531 |
| GreenSky | 662 | 190 | 8,689 | 1,025 |
| News | 42,112 | 75 | 2,115,322 | 1,314 |
| Political Science | 357 | 46 | 3,799 | 1,651 |
| Science | 33,831 | 1,716 | 980,724 | 4,506 |
| What's History | 161 | 71 | 2,462 | 821 |

- `feed_bookmarks.csv`. This file contains users who liked any of the collected feed generators. Each record contains three comma-separated values: the feed name (as per Table 3), the user ID, and the timestamp.

- `feed_post_likes.tar.gz`. This compressed folder contains data on likes to posts appearing in the feeds, one file per feed. Each record in the files contains the following information, in this order: the ID of the "liker", the ID of the post's author, the ID of the liked post, and the like timestamp;

- `scripts.tar.gz`. A collection of Python scripts, including the ones originally used to crawl the data and to perform experiments.

Our dataset abides by principles for FAIR data [27] since it is:

- *Findable*. We release our data on Zenodo, a service that stores data and assigns a DOI to the repository, ensuring findability. Moreover, we indexed it also on the SoBigData Research Infrastructure (URL: http://sobigdata.eu/), an EU-funded RI providing curated datasets and algorithms advocating open science;

- *Accessible*. Given the above, we ensure our data is freely accessible to anyone with an internet connection;

- *Interoperable*. All data is released in CSV or JSON files, enabling easy manipulation and analysis with most programming languages;

- *Reusable*. Our data collection process is transparent, and the shared data is appropriately described. Thus, it can be reused and inputted into many analytical pipelines.

## Ethics statement

The release of a large dataset of online interactions may raise significant ethical considerations that demand transparent handling. As stated by the service's Privacy Policy, *any information [a user] add[s] to [their] public profile and the information [they] post on the Bluesky App will be public* [28], and there is currently no option to turn a profile private. Therefore, all

information we collect is strictly public, including usernames, posts, and any attached metadata. Nonetheless, we strive to preserve privacy and anonymity for users in our sample. We have removed usernames from our dataset to mitigate privacy risks and replaced them with numerical IDs. We filtered out metadata that may univocally identify individuals or their content (e.g., post URIs) and aggregate temporal data at the minute level. The only data we collect regarding user profiles is the server they are registered to (information that is unlikely to provide any means of identifying individuals). Therefore, user bios, pictures, and registration dates—all potential identity markers—were not collected. These steps are detailed in the Data Processing section. Our dataset complies with Bluesky's Terms of service as well as with the latest European Union provisions in terms of data protection, particularly GDPR.

## Data analysis

In this Section, we describe the dataset in detail, providing insights into social topology, user activity and content. We also highlight potential applications of the dataset for social media mining.

### Social structure: Network topology and federation

**Followers network.** We model the system emerging from follower-followee relations as a directed graph $G = (V, E)$, such that $V = \{v_1, \ldots, v_n\}$ is the set of nodes/users, and $E = \{e_{ij}, \ldots, e_{km}\}$ is the set of edges/relations. Our snapshot of the Bluesky network counts 4,099,699 nodes connected by 14,458,1603 edges. A considerable fraction of these relations, $\sim 39\%$, are mutual. The distributions of in-degrees and out-degrees, representing the number of followers and followees for each account, follow a power law. Thus, few accounts hold most of the social capital, while the vast majority have only a few inward/outward connections. This characteristic is also shared by other online social networks such as Twitter/X and Facebook [29, 30]. The highest in-degree is 770,556 (i.e., of the most followed account), while the highest out-degree is 225,094 (i.e., of the account that follows others the most).

**Interaction networks.** Bluesky's post metadata allows modelling its social topology in different ways. Interaction networks can be built from the dataset leveraging replies, reposts, or quotes metadata according to the desired semantics and can possibly be multilayer [31, 32], weighted [33], and evolve in time [34, 35]. For instance, the conversation network —where nodes are users and directed edges $u \rightarrow v$ represent replies—resulting from the dataset is made up of 1.5M nodes connected by 23.4M edges. We find a remarkably high edge reciprocity of 57%, suggesting that users who receive a reply often reply back. Another interesting network is the one obtained by combining reposts and quotes (this is akin to retweet networks, a popular modelling choice for content/opinion diffusion studies on social media). The resulting network contains 1.4M nodes connected by 33.8M edges. The reciprocity rate is 8%, which is expected as most users typically share posts from a few hubs (e.g., news profiles and influencers). Apart from these hubs—some of which are reposted/quoted by more than 50,000 distinct users—we identify some account "boosters", i.e., accounts that share content from the same user thousands of times, effectively boosting her outreach. Indeed, by assigning edge weights based on (directed) interaction frequency, we find 140 unique node pairs interacting more than 1,000 times. Here, we have provided a brief description of some interaction layers, but future works could explore their joint topology, temporal dynamics [34, 35], as well as other structures such as mentions networks [36] and higher-order topologies [27, 37]. Moreover, by integrating interaction networks with follower relations, future studies could more adequately investigate content diffusion patterns on a comprehensive topology. Finally, the different semantic nuances of quotes and reposts potentially allow distinguishing between

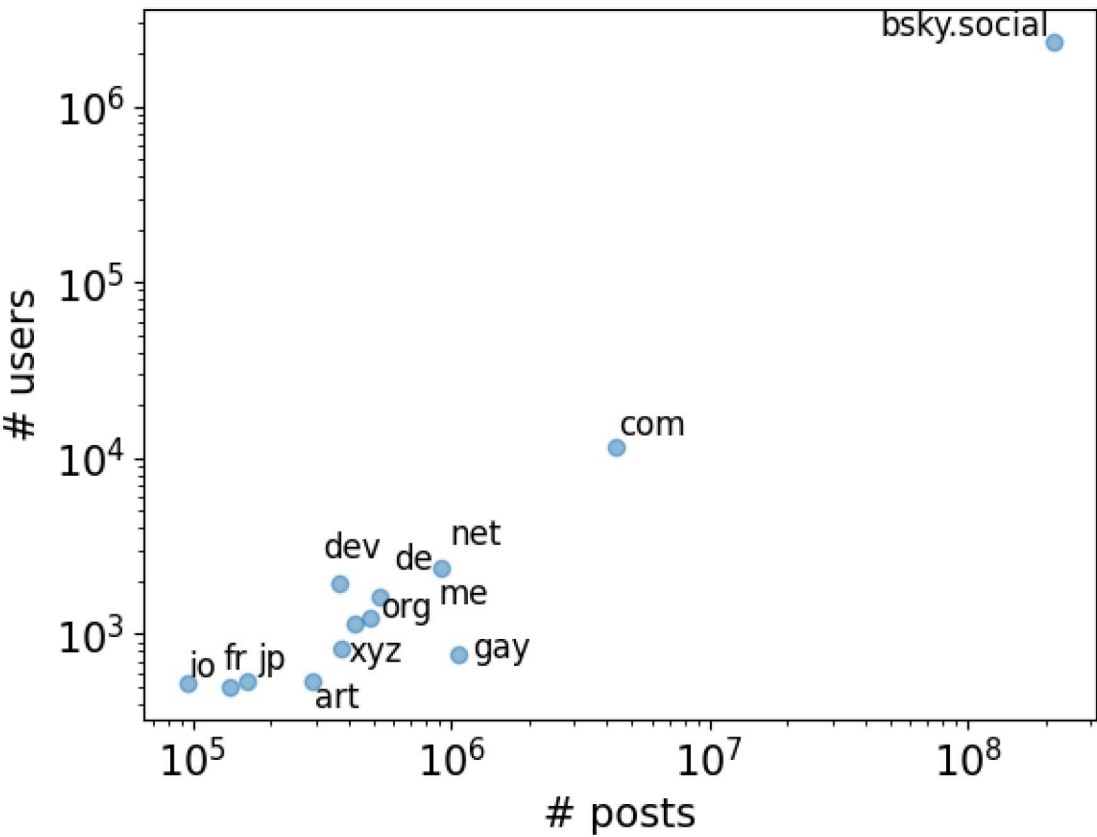

**Fig 2. Most populated and active instances.** Values are scaled logarithmically.

reposting as a means of endorsement and reposting as a means of criticizing, a well-known problem in content diffusion analysis [38].

**Federation.** The federated model is one of Bluesky's key characteristics. However, the platform only opened to federation on February 22nd, 2024. Before such a date, users could only register to the `bluesky.social` instance or create instances in a test sandbox network separated from the main one. Although it is impossible to tell whether a user has been registered to a server from the beginning or simply moved to a server at a later date, we argue that this is relatively unimportant since this does not affect what content a user is exposed to (see the protocol paper [15] for more details). Still, analyzing how users are distributed among instances can provide fundamental insights into the platform's structure and dynamics [39].

We display the instances with at least 500 members and 100,000 posts in Fig 2. Despite Bluesky's federation being relatively novel, the dataset contains posts from users registered to 6,181 servers. In the 26 days spanning from February 22nd to March 18th, $\sim 20M$ posts (8% of all posts) were shared on new servers, suggesting fast growth in the federation network. Aside from the default one, which is by far the most populated (98% of all users) and active (92% of all posts), other popular instances include common domains such as `.com` and `.net`, which are widely used by companies and news organizations. Some domains relate to specific countries/languages (`.de`, `.fr`, `.ca`), jobs and interests (`.dev`, `.art`) and other personal characteristics such as belonging to/supporting the LGBT+ community (`.gay`). In principle, this might allow topic-specific social studies on the platform. For instance, country/language

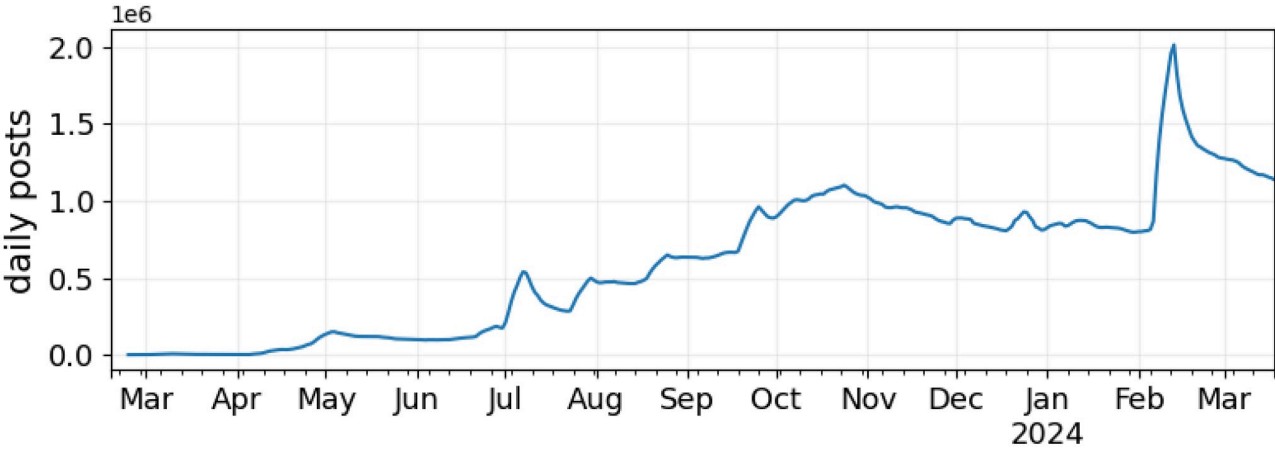

**Fig 3. Temporal trends of the number of average posts per day.**

domains could be used as proxies for coarse-grained geolocation data, which is not currently available in post metadata.

## Posting activity

Users show moderate engagement with the Bluesky platform. Out of the $\sim$4M users, nearly 2.4M (58%) shared at least one post. On average, these accounts shared 99 posts each ($\sigma$ = 717.63), with a median of 8. Of the 235M posts in our data, 63M (27%) are reposts, and 12M (5%) are quotes, indicating substantial content dissemination within the platform. Moreover, 20M discussion threads containing 88M replies can be identified.

Fig 3 shows the number of daily posts on the platform. Values are averaged via a seven-day rolling window to improve readability. Posts steadily increased from March through November, with some downward oscillations (e.g., around July-August). Activity on the platform begins to stabilize at $\sim$1M daily posts starting mid-October through February. A steep increase was registered after February 6th, when Bluesky's invite-only policy was lifted, effectively doubling the engagement on the platform. After that, activity decreased but remained higher than before the policy was lifted.

For a clearer picture of Bluesky's activity, we compute the daily average number of posts per active user. Formally, let $U$ be the set of users that posted at least once on day $d$, we compute:

$$\frac{\sum_u^U p_u}{|U|},  \tag{1}$$

where $p_u$ is the number of posts shared by user $u$ in $d$. The red curve in Fig 4 depicts this measure's trend. Once again, values are averaged via a seven-day rolling window. In this chart the light blue area outlines the interquartile range (IQR), i.e., where the middle 50% $p_u$ values are located; The blue curve outlines the average value within the IQR, providing a score less sensitive to outliers. During our observation period, the global average consistently falls beyond the IQR, highlighting the presence of very active users. We speculate that these might be (i) news organizations that continuously post updates and/or (ii) spammers and bots, as some accounts share up to 5K daily posts, which is unlikely for real accounts. The latter scenario also allows employing this dataset for bot detection [40] and coordinated behaviors analysis [5]. The IQR

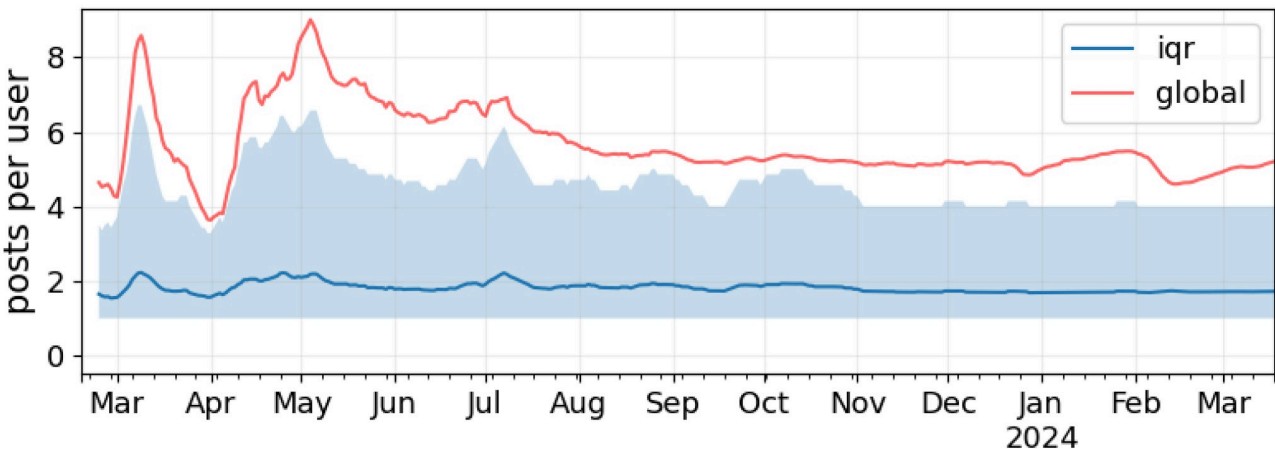

**Fig 4. Temporal trends of the average number of posts per active user.** The red line represents the average value, the blue area represents the interquartile range of daily posts per user, and the blue line represents the mean computed over the interquartile range.

average is mostly constant at around two posts per user. Notably, the February spike observed in Fig 3 does not affect these values.

Finally, to understand whether users are active for long or short periods of time, we turn to Fig 5, which shows the Cumulative Distribution Function (CDF) of days elapsed from each

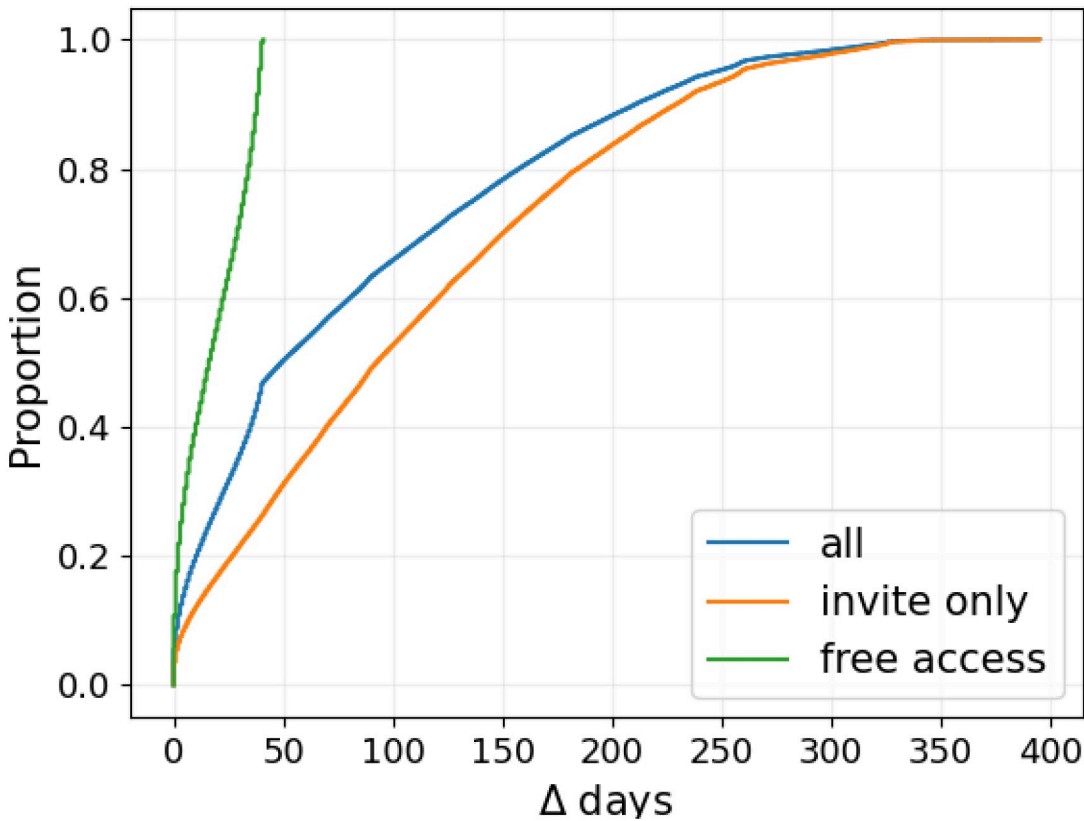

**Fig 5. Cumulative Distribution Function of the inter-event time from users' first to last posts (days).**

user's first to last posts. Globally, 50% of users were active for 50 days or longer, and 22% were active for 150 days or longer. To control for the effect of the invite-only removal, we additionally differentiate between *invite-only* (i.e., users whose first post is recorded before February 6th, 2024), and *free access* (i.e., users whose first post is recorded after the access policy was modified). *Invite-only* users have been mostly active for at least 75 days. Regarding the *free access* users, 40% were active for 25 days or more.

## Content analysis

**Languages.** In the following, we characterize posts based on their language metadata. When a user submits a post on Bluesky, she can select the language(s) that appear in the post via a dropdown menu. Users can choose up to three language tags since a post can be written in multiple languages. Language metadata was first added at the end of June 2023; thus, the languages of the earlier posts are unknown. In total, the dataset contains 227 unique language tags. The most frequent language is English, with over 132M posts (note that multilingual posts are counted once for each language). Japanese and German are also clusters of considerable size, respectively, 33.9M and 26M. The dataset also contains 5.2M posts with two language tags and 2.5M with three. Among the posts with two tags, we find English and Japanese (98K), English and Spanish (52K), and English and Portuguese (52K). Among the posts with three tags, we find English, Japanese, and Korean (233K), German, English, and French (103K), and English, Portuguese, and Spanish (87K). The wide variety of languages and the presence of multilingual posts make this dataset a valuable choice for multilingual studies on online social platforms.

**Feeds.** Feed generators are one of the main nuances of Bluesky. We have collected posts and metadata from 11 feed generators, shown in Table 3. The feeds in our sample cover a broad range of topics. Among these, some are related to socio-political issues (`#UkrainianView`, `GreenSky`), and sciences and academia (`Science`, `AcademicSky`, `Political Science`, `What's History`). Moreover, the `News` feed contains posts from verified news organizations. Other notable feeds relate to minorities/discriminated groups, such as `#Disability` and `BlackSky`. Among these, `BlackSky` has a peculiar opt-in mechanism for choosing whose posts to show. People who identify as black can ask the feed administrator to be added to the feed. Once they have been added, anything they post will be shown in the feed. Finally, the `BookSky` and `Game Dev` feeds refer to book recommendations and game development, presumably online spaces where users may look for advice/suggestions on these topics. To glance at the feeds' contents, the rightmost column in Table 4 shows the most frequent words in each feed. These were obtained via a standard text processing pipeline, including lowercasing, lemmatization, and removing punctuation, stopwords, numbers, emojis, and URLs. At a glance, most of these frequent words are somewhat related to the main topic of each feed. For instance, on `GreenSky`, terms refer to climate change, emissions, and non/renewable energy sources. This suggests that, despite their relatively small size, feeds may be used as effective proxies in topic-specific studies—in the same way as hashtags are on Twitter or subreddits on Reddit.

Information on when a user liked a feed generator may inform how much she is exposed to content about that topic. In principle, this may allow for the study of the effects of content exposure in a controlled way, i.e., by comparing activity before/after liking the feed. On a more global scale, feed liking information can outline patterns of increasing/decreasing popularity of certain topics, possibly influenced by real-world events. For instance, the `News` feed was liked the most in October 2023 (516 times), with a spike of 126 on October 21st. We hypothesize this may be tied to ongoing conflict in Gaza. The `Science` feed, instead, was

**Table 4. Most frequent words appearing in each feed.**

| name | words |
|---|---|
| #Disability | disability, disabled, work, covid, today, help, life, week, read, long |
| #UkrainianView | ukrainianview, russian, russia, ukraine, ukrainian, war, air, another, fuck, support |
| AcademicSky | academicsky, edusky, academia, highered, student, research, university, academic, psychscisky, bitly |
| BlackSky | black, love, today, work, feel, white, life, woman, shit, post |
| BookSky | book, booksky, read, reading, review, author, story, today, finished, horror |
| Game Dev | game, gamedev, dev, design, art, indiedev, indiegame, screenshotsaturday, work, whatagamedevlookslike |
| GreenSky | climate, energy, change, carbon, power, emission, fuel, fossil, work, global |
| News | news, ukraine, russian, state, russia, president, trump, please, feed, orgs |
| Political Science | polisky, feed, polisci, political, list, book, student, post, science, gendersky |
| Science | science, feed, post, today, paper, please, research, data, work, study |
| What's History | history, skystorians, book, american, war, eel, historical, japanese, woman, read |

mostly liked in the summertime, receiving nearly 1,200 likes in July, Moreover, like data on posts can give more fine-grained information about the popularity of specific topics, posts, or users. For instance, in the Science feed, the most liked post has 250,600 likes. It received the most likes when it was posted, namely 15K on February 10th, 2024, and 17K the next day. After that, its popularity decreased, although it received 1K to 4K daily likes up until mid-April.

**Sentiment and topics.** Sentiment can be an interesting indicator of the overall emotional atmosphere on the platform. Out of the annotated English posts, 39M (32%) are positive, 32M (27%) are negative, and 50M (41%) are neutral. Daily sentiment rates are shown in Fig 6. Note

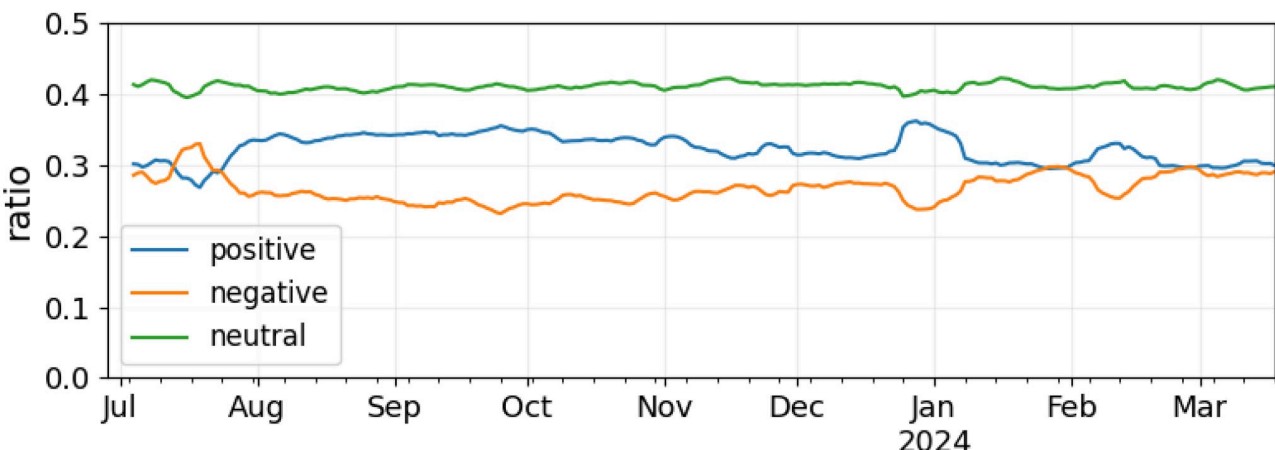

**Fig 6. Temporal trends of English post sentiment.**

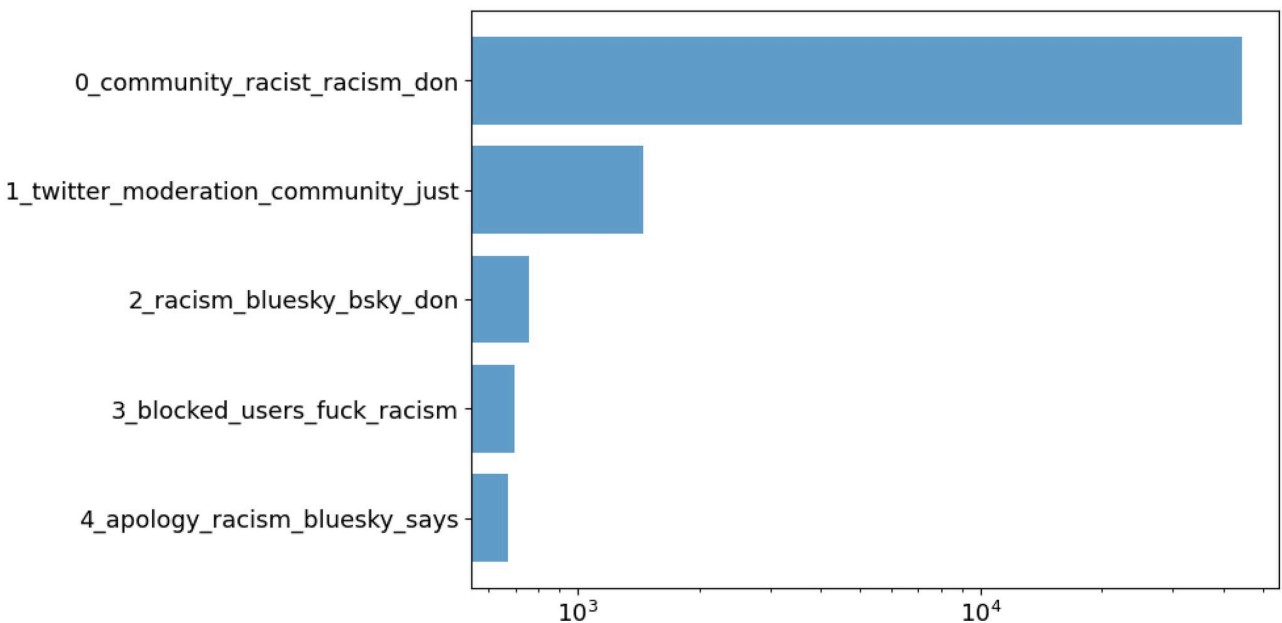

**Fig 7. Count of the five largest document clusters as identified by BERTopic on negative posts issued within July 13-15th.** Values on the x-axis are scaled logarithmically.

that posts before July do not have any language metadata and are thus excluded from this analysis. Although trends are mostly flat, some interesting patterns can be observed. First, positive posts are relatively more than negative posts during most observations, highlighting a generally positive outlook within the community. A relative increase in positive posts can be seen around January 2024, likely due to the beginning of the new year and the associated sense of renewal and optimism. Moreover, the February influx of new users brought excitement and enthusiasm, leading to a temporary increase in positive posts. Another notable time window is July 13th to 15th. In this period, the platform is characterized by a relative increase in negative posts. To understand why, we leverage BERTopic, a neural topic modelling algorithm [41]. This model identifies topics by (i) constructing document embeddings, (ii) applying the UMAP algorithm for dimensionality reduction [42], and ultimately discovering document clusters via HDBSCAN [43]. We apply BERTopic to English posts published between July 13th and 15th (inclusive) and whose sentiment is negative. The model automatically identifies 40 clusters. The five largest ones are displayed in Fig 7 along with descriptive words for each topic. Although the topics are several, most documents belong to cluster 0, which refers to racism within the Bluesky online community. Other clusters, each with ∼1000 posts or less, seem to tackle the same issue from different perspectives. For instance, posts in topic 1 relate to content moderation, topic 3 mentions blocked users and topic 4 refers to apologies and racism. Thus, it is likely that the Bluesky community suffered from large-scale racist episodes, possibly concerning content moderation and/or platform administration. This hypothesis is confirmed by the online newspaper *TechCrunch*: on July 18th, an article reported a community backlash following Bluesky's failure to flag racial slurs in usernames [44]. At the same time, the platform allegedly removed many such words from its flagged words list. Looking back at Fig 3, this matter likely caused the sudden bump in posting activity in July. Future studies could investigate whether similar trends emerge in other languages as well.

## Conclusion

In this work we described the Bluesky Social Dataset, to the best of our knowledge, the first public dataset on the year-old decentralized social platform. Our dataset contains over 235M posts covering the full content history of more than 80% of all registered users. Longitudinal interaction data is also made available, including follow, reply, repost, and quote interactions. We also exploit the peculiar feature of *feed generators* and collect all posts served from the most popular recommenders on the platform, together with information on users who subscribed to these feeds. The data is hosted on Zenodo [25], along with the code used to collect it, clean it, and reproduce the plots. Overall, we observe good engagement patterns on the platform. Users actively participate across a variety of topic-specific environments, including feeds and instances tailored to particular interests, demographics, and languages. Additionally, we notice that global user behaviors emerge in response to real-world events, leading to spikes in activity and shifts in discourse as users react to current happenings. Our contribution paves the way toward a deeper understanding of digital interactions and dynamics on niche and/or decentralized platforms. Moreover, combining multiple layers of social topology with information extracted from user-generated content might lead to new insights into polluting dynamics such as d/misinformation, opinion polarization, and political segregation.

## Usage notes

The data is hosted on Zenodo [25]. The platform allows hosting datasets up to 50GB and 100 files. We provide the data in compressed formats to comply with file size and amount limitations. To re-run the analysis on posts, networks, and feeds, it is necessary to decompress the corresponding archive(s).

## Code availability

All code related to generating, processing, and describing the dataset is released alongside it. The code requires Python 3.8 or higher and is mostly based on standard libraries and popular data science packages (e.g., numpy, pandas, matplotlib). The scripts for data collection also require `atproto` v0.0.46, the official Python wrapper for Bluesky Social API. Please refer to the API documentation for further details [22]. Before running data collection scripts, users should also set a `USERNAME` and `PASSWORD` environment variables with their Bluesky credentials, as some methods may require authentication.

## Author Contributions

**Data curation:** Andrea Failla.

**Funding acquisition:** Giulio Rossetti.

**Investigation:** Andrea Failla.

**Methodology:** Andrea Failla.

**Software:** Andrea Failla.

**Supervision:** Giulio Rossetti.

**Visualization:** Andrea Failla.

**Writing – original draft:** Andrea Failla.

**Writing – review & editing:** Giulio Rossetti.

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
