## [Decision Letter · Decision Letter 0]

26 Aug 2024

PONE-D-24-24931“I’m in the Bluesky Tonight”: Insights from a Year Worth of Social DataPLOS ONE

Dear Dr. Failla,

Thank you for submitting your manuscript to PLOS ONE. After careful consideration, we feel that it has merit but does not fully meet PLOS ONE’s publication criteria as it currently stands. Therefore, we invite you to submit a revised version of the manuscript that addresses the points raised during the review process.

We look forward to receiving your revised manuscript.

Kind regards,

Fabio Saracco

Academic Editor

PLOS ONE

2. In your Methods section, please include additional information about your dataset and ensure that you have included a statement specifying whether the collection and analysis method complied with the terms and conditions for the source of the data.

“This work is supported by (i) the European Union – Horizon 2020 Program under the scheme “INFRAIA-01-2018-2019 – Integrating Activities for Advanced Communities”, Grant Agreement n.871042, ''SoBigData++: European Integrated Infrastructure for Social Mining and Big Data Analytics" (\\url{http://www.sobigdata.eu}); (ii) SoBigData.it which receives funding from the European Union – NextGenerationEU – National Recovery and Resilience Plan (Piano Nazionale di Ripresa e Resilienza, PNRR) – Project: ''SoBigData.it – Strengthening the Italian RI for Social Mining and Big Data Analytics" – Prot. IR0000013 – Avviso n. 3264 del 28/12/2021; (iii) EU NextGenerationEU programme under the funding schemes PNRR-PE-AI FAIR (Future Artificial Intelligence Research).”

“All code related to generating, processing, and describing the dataset is released alongside it. The code requires Python 3.8 or higher and is mostly based on standard libraries and popular data science packages (e.g., numpy, pandas, matplotlib). The scripts for data collection also require atproto v0.0.46, the official Python wrapper for Bluesky Social API. Please refer to the API documentation for further details [22]. Before running data collection scripts, users should also set a USERNAME and PASSWORD environment variables with their Bluesky credentials, as some methods may require authentication.”

“This work is supported by (i) the European Union – Horizon 2020 Program under the scheme “INFRAIA-01-2018-2019 – Integrating Activities for Advanced Communities”, Grant Agreement n.871042, ''SoBigData++: European Integrated Infrastructure for Social Mining and Big Data Analytics" (\\url{http://www.sobigdata.eu}); (ii) SoBigData.it which receives funding from the European Union – NextGenerationEU – National Recovery and Resilience Plan (Piano Nazionale di Ripresa e Resilienza, PNRR) – Project: ''SoBigData.it – Strengthening the Italian RI for Social Mining and Big Data Analytics" – Prot. IR0000013 – Avviso n. 3264 del 28/12/2021; (iii) EU NextGenerationEU programme under the funding schemes PNRR-PE-AI FAIR (Future Artificial Intelligence Research).”

5. We note that Figure 1 in your submission contain copyrighted images. All PLOS content is published under the Creative Commons Attribution License (CC BY 4.0), which means that the manuscript, images, and Supporting Information files will be freely available online, and any third party is permitted to access, download, copy, distribute, and use these materials in any way, even commercially, with proper attribution. For more information, see our copyright guidelines: http://journals.plos.org/plosone/s/licenses-and-copyright.

Additional Editor Comments:

Dear authors,

The reviewers provided extremely positive reports. There just a few small points to be clarified in order to improve the readability of the paper.

Reviewers' comments:

Reviewer's Responses to Questions

**Comments to the Author**

1. Is the manuscript technically sound, and do the data support the conclusions?

Reviewer #1: Yes

Reviewer #2: Yes

2. Has the statistical analysis been performed appropriately and rigorously? 

Reviewer #1: N/A

Reviewer #2: Yes

3. Have the authors made all data underlying the findings in their manuscript fully available?

Reviewer #1: Yes

Reviewer #2: Yes

4. Is the manuscript presented in an intelligible fashion and written in standard English?

Reviewer #1: Yes

Reviewer #2: Yes

5. Review Comments to the Author

Reviewer #1: With this publication, the authors provide the community with a dataset of approximately 235 million posts collected from the new social network 'Bluesky'. The data gathered cover about 80% of the accounts on the platform in terms of user involvement.

PROS/CONS

(+) The study presents an original dataset; no other Bluesky dataset with similar coverage appears to be available in the literature.

(+) The dataset represents an alternative approach to an important recent problem, i.e., the difficulty of accessing social media data in the "post-API era." Furthermore, the dataset represents the Bluesky social media at its initial stage.

(-) I believe that some aspects of the data collection could be better clarified by the authors.

This reviewer believes that the paper is well-structured and clear to read. Moreover, although I believe some aspects could be clarified further, the authors describe the data collection process allowing for the reproducibility of the approach. Also, given the recent developments and the increasing difficulty in accessing social data for research purposes ("post-API era"), I believe that the contribution is important for the community.

Here are two aspects regarding the data collection that I think should be clarified (I am confident that the authors will be able to manage them):

- The "relational" data and the posts might not be aligned? I will try to explain better. If I understand correctly, in the first phase, the relational data (followers of various accounts) are collected, and subsequently, all the posts published on the timelines of the collected users. Potentially, some follower/followee relationships might be added or removed during the collection of the posts. I would ask the authors to discuss whether this might represent a limitation in some studies and how to manage it (e.g., if it were a problem, to have "relational" data and posts aligned, one might consider only the posts published up until the start of data collection for the followers?)

- Text from row 98 to 101: I would like to know if the authors have encountered the same problem for replies. In other words, how did they separate replies from reposts? Does the metadata provided by Bluesky already provide this information?

Reviewer #2: The paper presents a very rich dataset of bluesky users, relationships and interactions, made available by the authors together with the software used to collect It and preprocess it.

In addition, the authors present an exploratory analysis of the dataset, that provides a first picture of this OSN, its structure, dinamics and topics of discussion. Leveraging on the analysis, the authors discuss possibile ways to make use of these data to advance open issues in computational social sciences.

I found the paper very easy and pleasant to read. While the analysis is not very deep, the dataset itself may be a very useful tool for other researchers, and the paper successfully delineates possibile use cases and research directions.

I only have 2 very small commenta:

- in line 43, the authors missed an upper case after a dot

- I would rename the "Results" section to something like "Data analysis"

6. PLOS authors have the option to publish the peer review history of their article (what does this mean?). If published, this will include your full peer review and any attached files.

Reviewer #1: No

Reviewer #2: No

---

## [Author Response · Author response to Decision Letter 0]

27 Aug 2024

We would like to thank the editor and the reviewers for their time and valuable feedback. Their suggestions and comments have significantly contributed to enhancing the quality of our paper. Below, we provide a detailed, point-by-point response to each report.

Academic Editor.

Q. 1. Please ensure that your manuscript meets PLOS ONE's style requirements, including those for file naming. The PLOS ONE style templates can be found at 

A. We have reviewed the style requirements and ensured compliance.

Q. 2. In your Methods section, please include additional information about your dataset and ensure that you have included a statement specifying whether the collection and analysis method complied with the terms and conditions for the source of the data. 

A. We moved the Ethics statement to the Materials and Methods section. The statement now specifies that the collection and analysis methods comply with the platform's terms of service. 

Q. 5. We note that Figure 1 in your submission contain copyrighted images. All PLOS content is published under the Creative Commons Attribution License (CC BY 4.0), which means that the manuscript, images, and Supporting Information files will be freely available online, and any third party is permitted to access, download, copy, distribute, and use these materials in any way, even commercially, with proper attribution. For more information, see our copyright guidelines: http://journals.plos.org/plosone/s/licenses-and-copyright.

We require you to either (1) present written permission from the copyright holder to publish these figures specifically under the CC BY 4.0 license, or (2) remove the figures from your submission 

A. We updated the figure removing potentially copyrighted material. Moreover, the caption now clarifies the image source (screenshot taken by the author from their own device). 

Q. 3-4. 

These changes were addressed in the cover letter as requested.

Reviewer 1. 

Q. The "relational" data and the posts might not be aligned? I will try to explain better. If I understand correctly, in the first phase, the relational data (followers of various accounts) are collected, and subsequently, all the posts published on the timelines of the collected users. Potentially, some follower/followee relationships might be added or removed during the collection of the posts. I would ask the authors to discuss whether this might represent a limitation in some studies and how to manage it (e.g., if it were a problem, to have "relational" data and posts aligned, one might consider only the posts published up until the start of data collection for the followers?)

A. As the reviewer notes, the “photograph” we take of a social media platform is not instantaneous but gradual, requiring time to be fully captured. Consequently, changes such as relation/post addition/removal may be overlooked if they happen after the corresponding area of the network is parsed. Thus, considering that a degree of information loss is inevitable at such a scale, we took some measures to limit its repercussions. Specifically, we last parsed the follower network on March 19th (this process started and ended on the same day), and released posts until March 18th (i.e., immediately before starting to parse follower relations). By doing so, we ensure that the misalignment is kept to a minimum (about 24 hours). We clarified this aspect in lines 135-139 by highlighting the advantage of this filtering process.

Q. Text from row 98 to 101: I would like to know if the authors have encountered the same problem for replies. In other words, how did they separate replies from reposts? Does the metadata provided by Bluesky already provide this information? 

A. Bluesky references the post to which a reply is linked within post metadata. As the reviewer noted, we did not specify this aspect in the previous version of the manuscript. In the attached version, we state that replies and quotes have this relevant information as metadata, as opposed to reposts (lines 99-101). 

Reviewer 2. 

Q. in line 43, the authors missed an upper case after a dot

A. The typo is now fixed 

Q. I would rename the "Results" section to something like "Data analysis" 

A. The Results section was renamed to Data Analysis

---

## [Editor Report · Decision Letter 1]

29 Aug 2024

“I’m in the Bluesky Tonight”: Insights from a Year Worth of Social Data

PONE-D-24-24931R1

Dear Dr. Failla,

We’re pleased to inform you that your manuscript has been judged scientifically suitable for publication and will be formally accepted for publication once it meets all outstanding technical requirements.

Kind regards,

Fabio Saracco

Academic Editor

PLOS ONE

Additional Editor Comments (optional):

Dear authors,

since all the requests raised by the referees were properly addressed, I believe that the manuscript is ready for publication. Congratulations!

Best,

Fabio Saracco
---

## [Editor Report · Acceptance letter]

3 Sep 2024

PONE-D-24-24931R1 

PLOS ONE

Dear Dr. Failla, 

I'm pleased to inform you that your manuscript has been deemed suitable for publication in PLOS ONE. Congratulations! Your manuscript is now being handed over to our production team.

Kind regards, 

on behalf of

Dr. Fabio Saracco 

Academic Editor

PLOS ONE